# Solvent Accessibility of Coronaviridae Spike Proteins through the Lens of Information Gain

**Sarwan Ali** [ID], **Babatunde Bello** [ID] and **Murray Patterson** *[ID]

Department of Computer Science, Georgia State University, Atlanta, GA 30303, USA;
sali85@student.gsu.edu (S.A.)
* Correspondence: mpatterson30@gsu.edu

**Abstract:** The COVID-19 pandemic, caused by the coronavirus SARS-CoV-2, has generated a renewed interest in the larger family of Coronaviridae, which causes a variety of different respiratory infections in a variety of different hosts. Understanding the mechanisms behind the ability of a family of viruses to spill over into different hosts is an ongoing study. In this work, we studied the relationship between specific amino acid sites and the solvent accessibility of the surface (or spike) protein of different Coronaviridae. Since host specificity hinges on the portion(s) of the protein that interfaces with the host cell membrane, there could be a relationship between information gain in specific amino acid sites and solvent accessibility. We found a connection between sites with high information gain and solvent accessibility within several major subgenera of Coronaviridae. Such a connection could be used to study other lesser-known families of viruses, which is desirable because information gain is much easier to compute when the number of sequences is large, as we show. Finally, we produced a visualization of the sequences within each major subgenus and discussed several regions of interest, as well as focused on some pairs of Coronaviridae hosts of interest.

**Keywords:** Coronaviridae; information gain; spike protein structure; solvent accessibility

## 1. Introduction

The viruses of the family Coronaviridae are so-called because of their peculiar crown-shaped surface (or spike) protein—see Figure 1 for the genomic structure of a coronavirus, which includes the spike region. Coronaviruses cause a variety of known respiratory infections [1], including SARS (SARS-CoV), Middle-Eastern Respiratory Syndrome (MERS-CoV), and, of course, SARS-CoV-2, which is responsible for the ongoing COVID-19 global pandemic. Many of the Coronaviridae affect a wide variety of hosts, and often spill over to other hosts [2]. SARS-CoV was believed to have jumped to humans via palm civets and horseshoe bats [3], MERS-CoV from dromedary camels [4], whereas SARS-CoV-2 likely originated from bats [5].

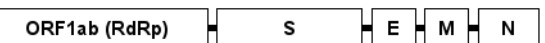

**Figure 1.** The genome of a coronavirus ranges from 26–32 kb in length [6], and codes for two non-structural and four structural proteins. The non-structural proteins are coded by ORF1ab, which contains the RNA-dependent RNA polymerase region (RdRp). The structural proteins include spike (S), envelope (E), membrane (M), and nucleocapsid (N). The S gene region encodes the spike protein, which is responsible for attaching the virus to receptors on the host cell membrane.

Our understanding of the mechanisms that could allow different viruses to spill over from one host to another is continuously updating as the viruses evolve and we gather more sequencing data. Here, we considered *information gain* (IG) [7] for explaining host specificity, something that is convenient in light of the constant accumulation of new

sequencing data, since the IG computation only requires a multiple alignment of a set of sequences.

**Definition 1** (Information Gain). *Information gain is a measure of how much information a feature provides about a target class [7]. Given a protein sequence, the IG of an amino acid site in terms of a host is the following:*

$$IG(host, site) = H(host) - H(host \mid site) \qquad (1)$$

*where*

$$H(C) = \sum_{i \in C} -p_i \log p_i$$

*where H(C) is the entropy of category C, the notation C represents the host/site, and $p_i$ is the probability of element i of category C.*

Intuitively, the information gain of a given amino acid site is how much information this site provides for deciding the host, e.g., whether the sequence is of a virus that infects bats, humans, or camels. Such a measure, while simple to compute, has the potential to capture important trends when the number of sequences analyzed is sufficiently large.

Since it is the portion(s) of the surface protein responsible for fusing to the host cell membrane that determines host specificity [8], one might expect information gain to be enriched for sites corresponding to solvent-exposed extracellular residues. For this reason, we performed solvent accessibility prediction for sequences in several different major subgenera (Sarbecovirus, Merbecovirus, etc.), and then demonstrated higher IG in solvent-exposed sites on the spike protein sequence.

The Coronaviridae are a family of enveloped, positive-sense single-stranded RNA viruses that can cause respiratory [9], gastrointestinal [10], liver [11], and neurological diseases [12] in humans and animals. The viral envelope, which is composed of a lipid bi-layer, is essential for survival and replication [13], as it protects the viral genome and mediates interactions with host cells. The solvent exposure of the virus determines the stability of the viral envelope and affects its interactions with host cells and the environment [14]. The spike proteins on the surface of the envelope are particularly sensitive to solvent exposure [15], and changes in solvent exposure can cause structural changes in these proteins, which can affect their function [16]. Therefore, the solvent exposure of the Coronaviridae is an important factor to consider when studying the biology and pathogenesis of these viruses. The advantages of studying solvent exposure in the spike protein of Coronaviridae include

- *Understanding protein stability:* Solvent exposure can provide insights into the stability and conformational changes in the spike protein, which are critical for its function and viral infectivity [17].
- *Drug discovery*: Knowledge of the solvent exposure of the spike protein can aid in the discovery of drugs that target the virus by disrupting the protein's stability or function [18].
- *Vaccine design*: Studying solvent exposure can inform the design of vaccines that elicit an immune response against the spike protein, which is a crucial target for neutralizing antibodies [19].
- *Mechanism of action*: Understanding the solvent exposure of the spike protein can provide insights into the mechanism of action of the virus and how it enters host cells [16].
- *Evolution*: Studying solvent exposure can also provide information on how the virus evolves and adapts to changing environments, which can inform the development of strategies to control the spread of the virus [17].

Recently, as the amount of sequencing data becomes large enough, some efforts have been made to study families of viruses using machine learning approaches. These

methods apply some form of classification [20] or clustering [21]. One way to improve the performance of these machine learning methods is to make use of biological domain knowledge [22]. Using statistical analysis is another method that could be used to improve the performance of underlying machine learning models (as suggested by Ali et al. in [20]). In this work, we studied the relationship between specific amino acid sites (within MSA) and the solvent accessibility of the surface (or spike) protein of different Coronaviridae. While we applied this study to the family Coronaviridae as a proof of concept, one could apply this to newer families of lesser-known viruses in the hopes of better understanding how protein structure affects their host specificity. Our contributions in this paper are the following:

1.  We used information gain (IG) to evaluate the importance of amino acid sites of the spike sequence, which could be used to study the behavior of the virus.
2.  Obtaining structural information about the spike protein, namely solvent accessibility, we evaluated if there is any relationship between this and IG.
3.  We show that there is such a connection between IG and solvent accessibility, the latter requiring much more time to compute.
4.  We performed several case studies to show the biological relevance of the relationship between solvent accessibility and IG.

The rest of the paper is organized as follows. Section 2 contains the detail on the data collection and statistics. Section 3 contains the study that shows a connection between information gain and solvent accessibility. Section 4 provides some case studies on regions and pairs of hosts of interest. Finally, we conclude our paper in Section 5.

## 2. Data Collection and Statistics

In this section, we first discuss details on the collection of the sequence data using different sources. We then provide some descriptive statistics of this data, and some visualization of the data using the t-distributed stochastic neighborhood embedding (t-SNE) method.

### 2.1. Data Collection

**Virus Pathogen Resource (ViPR):**

Data were obtained from ViPR (https://www.viprbrc.org) on 9 February 2022. We downloaded all 1,441,773 unaligned complete S protein sequences of the family Coronaviridae for host "All". Because this database is overwhelmed with SARS-CoV-2 sequences due to the COVID-19 pandemic, and the ViPR search utility is such that hosts can only be specified but not filtered (as far as we know), we downloaded all 1,439,557 such sequences for all hosts and the dataset for host "human" was excluded, resulting in a remaining 2216 sequences. Note that some of the sequences with no host label (according to ViPR) might still pertain to a human host. We then queried the National Center for Biotechnology Information (NCBI) resource (since the sequences from ViPR are annotated with NCBI IDs) to obtain the taxonomic and host information of these 2216 sequences. Of these, 14 sequences were from the Piscanivirinae, Serpentovirinae, and Torovirinae subfamilies, which were previously part of the Coronaviridae family but were re-classed to the Tobaniviridae family in 2018 [23]. We removed these 14, resulting in a remaining 2202 sequences, which are distributed across different taxa as given in Table 1.

**Table 1.** Genus/subgenus distribution over 2022 sequences for all hosts except for "Human" downloaded from ViPR. The "CoV" is short for "coronavirus".

| Genus | Subgenus | No. Sequences | Genus | Subgenus | No. Sequences |
|---|---|---|---|---|---|
| AlphaCoV | Colacovirus | 3 | BetaCoV | Embecovirus | 175 |
| AlphaCoV | Decacovirus | 11 | BetaCoV | Hibecovirus | 1 |
| AlphaCoV | Duvinacovirus | 33 | BetaCoV | Merbecovirus | 289 |
| AlphaCoV | Luchacovirus | 4 | BetaCoV | Nobecovirus | 11 |
| AlphaCoV | Minacovirus | 6 | BetaCoV | Sarbecovirus | 614 |
| AlphaCoV | Minunacovirus | 7 | BetaCoV | unknown | 11 |
| AlphaCoV | Nyctacovirus | 5 | DeltaCoV | Andecovirus | 2 |
| AlphaCoV | Pedacovirus | 370 | DeltaCoV | Buldecovirus | 147 |
| AlphaCoV | Rhinacovirus | 56 | DeltaCoV | Herdecovirus | 2 |
| AlphaCoV | Setracovirus | 7 | DeltaCoV | unknown | 4 |
| AlphaCoV | Tegacovirus | 51 | GammaCoV | Cegacovirus | 7 |
| AlphaCoV | unknown | 23 | GammaCoV | Igacovirus | 344 |
| unknown | unknown | 19 | - | - | - |

**Global Initiative on Sharing All Influenza Data (GISAID):**

Data were also obtained from GISAID (https://www.gisaid.org) on 9 February 2022. We downloaded all 7,940,305 unaligned S (spike) protein sequences of SARS-CoV-2 from GISAID. We then filtered out the 7,935,111 sequences tagged with host "Human" (or "Hombre"), resulting in a remaining 5194 sequences for all hosts except for "Human". Since there was no way to restrict the search to complete sequences (like with ViPR, or as far as we know), we removed 59 sequences that were more than three standard deviations ($\sigma = 115.38$) shorter than the mean ($\mu = 1259.27$) sequence length, resulting in a remaining 5135 sequences with lengths ranging from 1128 to 1276. Since these were all SARS-CoV-2 sequences, they were all of the subgenus Sarbecovirus (genus Betacoronavirus).

**Data Integration:**

We then combined these two datasets into a single dataset of $2022 + 5135 = 7337$ sequences, and then added 2663 randomly selected sequences from the 1,439,557 sequences for host "Human" downloaded from ViPR (above) for an even 10,000 sequences. All but one of these 2663 sequences were of the subgenus Sarbecovirus, that one being of subgenus Embecovirus (genus Betacoronavirus). In the end, the "major" subgenera and hosts were considered, those represented by more than 100 sequences, and are depicted in Tables 2 and 3.

**Data Processing:**

For each of the sets of sequences, represented by the subgenera and hosts, we performed a multiple alignment with Mafft ( https://mafft.cbrc.jp/alignment/software/) (accessed on 9 February 2022) using default parameters. We then removed from this alignment any position (column) containing only X and - , since they are a "wildcard" character and a gap, respectively. This is preferable to removing all X characters from the sequences beforehand, because it allows the aligner to decide what each X means, rather than making an irreversible decision. Since we are only interested in host specificity (not subgenus specificity), we computed the information gain of each site of the alignment for each subgenus of Table 2 only. Note that, while some subgenera have many more sequences than others (Table 2). The Host-based grouping of sequences is also reported in Table 3. The length statistics for different subgenera and hosts are reported in Table 4 and Table 5, respectively. In Table 2, most notable subgenus is Sarbecovirus, we studied the host specificity within each subgenus, independent of the others. Still, what this means is that the results will be more reliable for subgenera with a larger number of sequences—for example, in Table 6, Sarbecovirus has a notably stronger correlation value and smaller *p*-value—something we discuss more in Section 3 below. The information of the spike protein

was predicted using the SCRATCH Protein Predictor by taking protein sequences as the input (http://scratch.proteomics.ics.uci.edu/) (accessed on 9 February 2022), classifying the sites as being solvent-exposed on a scale of 1 to 20. We provide more statistics and visualization of these data in the subsequent sections.

**Table 2.** Subgenera-based grouping of sequences.

| Subgenus | No. Sequences |
|---|---|
| Buldecovirus | 147 |
| Embecovirus | 176 |
| Igacovirus | 344 |
| Merbecovirus | 289 |
| Pedacovirus | 370 |
| Sarbecovirus | 8411 |

**Table 3.** Hosts-based grouping of sequences.

| Host | No. Sequences |
|---|---|
| Bat | 233 |
| Camel | 290 |
| Cat | 292 |
| Chicken | 321 |
| Deer | 147 |
| Environment | 3423 |
| Human | 2927 |
| Pig | 553 |
| Weasel | 1195 |

**Table 4.** Length statistics, including average length, standard deviation length, and alignment length for the major subgenera.

| Subgenus | No. Seq. | Avg.Len. | SD | Align. Len. |
|---|---|---|---|---|
| Buldecovirus | 147 | 1163.10 | 12.49 | 1312 |
| Embecovirus | 176 | 1358.23 | 15.39 | 1445 |
| Merbecovirus | 289 | 1352.58 | 3.09 | 1403 |
| Igacovirus | 344 | 1164.93 | 7.31 | 1442 |
| Pedacovirus | 370 | 1382.84 | 17.72 | 1462 |
| Sarbecovirus | 8411 | 1270.90 | 4.15 | 1398 |

**Table 5.** Length statistics including average length, standard deviation length, and alignment length for the major hosts.

| Host | No.Seq. | Avg.Len. | SD | Align.Len. |
|---|---|---|---|---|
| Bat | 233 | 1282.06 | 66.15 | 2076 |
| Camel | 290 | 1335.49 | 54.46 | 1624 |
| Cat | 292 | 1282.86 | 44.02 | 1628 |
| Chicken | 321 | 1163.93 | 3.58 | 1199 |
| Deer | 147 | 1271.14 | 11.93 | 1276 |
| Environment | 3423 | 1271.33 | 1.26 | 1365 |
| Human | 2927 | 1272.57 | 11.56 | 1847 |
| Pig | 553 | 1310.72 | 109.15 | 1715 |
| Weasel | 1195 | 1272.85 | 11.96 | 1622 |

**Table 6.** Rank correlation comparisons of information gain and solvent accessibility.

| Subgenus | Spearman Rank Correlation | | Kendall Rank Correlation | |
|---|---|---|---|---|
| | Correlation Value | *p*-Value | Correlation Value | *p*-Value |
| Buldecovirus | 0.3319 | $3.996671379687281 \times 10^{-35}$ | 0.2400 | $1.9838872978121717 \times 10^{-35}$ |
| Igacovirus | 0.2245 | $5.60950280481785 \times 10^{-18}$ | 0.1558 | $1.2834910986811683 \times 10^{-17}$ |
| Embecovirus | 0.3382 | $5.330451027010657 \times 10^{-40}$ | 0.2409 | $9.358899326269566 \times 10^{-39}$ |
| Merbecovirus | 0.2995 | $1.7944803412415015 \times 10^{-30}$ | 0.2114 | $4.157343061564072 \times 10^{-29}$ |
| Pedacovirus | 0.2206 | $1.4050087772378643 \times 10^{-17}$ | 0.1626 | $3.840562973426854 \times 10^{-17}$ |
| Sarbecovirus | 0.6109 | $6.314866462884465 \times 10^{-168}$ | 0.4464 | $7.402190248518821 \times 10^{-160}$ |

## 2.2. Dataset Statistics

We first give some statistics on the average sequence length and standard deviation (SD) of the length distribution for the major subgenera and hosts in Tables 4 and 5, respectively. The last column of both tables shows the length of the alignment template. Note that this length tends to grow larger than the average length as both the number of sequences and standard deviation grow, which is to be expected.

The distribution of hosts within each subgenus is given in Figure 2. Our pre-processed dataset is available online (https://github.com/sarwanpasha/Comparative_Genomics) (accessed on 9 February 2022). Note that some subgenera, in particular Embecovirus and Sarbecovirus, have a much more widespread diversity of hosts than others. We note that host label (Table 3 and Figure 2) are discrete, when, in reality, the sequences exhibit a spectrum of host specificity; here, we just chose the most common host for each sequence from the records (NCBI, etc.) above. With a large enough number of sequences (e.g., Sarbecovirus, Table 2), overall trends in host specificity should have a clear signal.

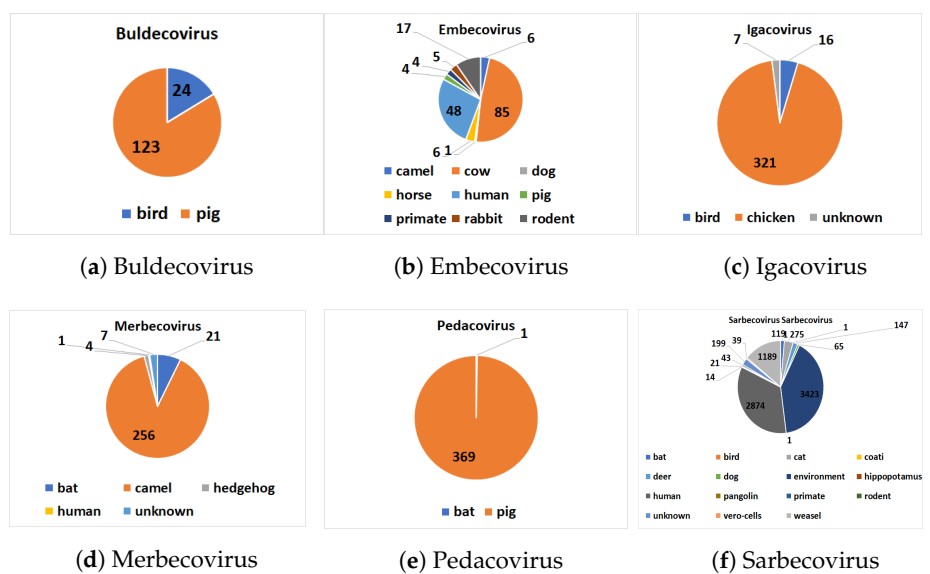

**Figure 2.** Host distribution with each major subgenus.

## 2.3. Visualization with t-SNE

In order to visually evaluate if there is any (hidden) clustering in the sequences, we used the t-distributed stochastic neighbor embedding (t-SNE) [24] method. The t-SNE method maps input data to 2D real vectors, which can then be visualized as a scatter plot. Since t-SNE takes numerical vectors as input, we obtained a numerical representation of the sequences of each subgenus using a feature vector generation called Spike2Vec [20].

Given some sequence $\sigma$ on alphabet $\Sigma$, Spike2Vec generates substrings (also called mers) of length $k$, i.e., $k$-mers. In $\Sigma$, we have the following 20 characters (amino acids): "ACDEFGHIKLMNPQRSTVWY". From a sequence, $k$-mers are generated by applying a sliding window of size $k$ over the sequences. Given a sequence of length $N$, the total number of $k$-mers that could be generated is $N - k + 1$.

After generating the *k*-mers, Spike2Vec creates a feature vector $\Phi$ (a frequency vector), which contains the frequency (count) of each *k*-mer occurring in the sequence. The length of the feature vector $\Phi_k(\sigma)$ is $|\Sigma|^k$. Since we worked with spike protein (amino acid) sequences and took $k = 3$ in our experiments, the feature vector length that we used was $20^3 = 8000$. This feature vector can be used as the input for t-SNE plots. The t-SNE plots of the sequences of each of the six major subgenera, colored by host, are given in Figure 3. In Embecovirus, we note that the two major groups, Cow and Human, are well separated. In Merbecovirus, there is some grouping for the Camel host. Interestingly, in the Pedacovirus, there seem to be two distinct groups of sequences that both affect the Pig (or Swine). Finally, in Sarbecovirus, there is some grouping of Humans, Weasel, and Bat. While these t-SNE plots capture sequence variability, this may or may not be related to information gain, since only a few such sites in the sequence might be important for specifying the host—something we explored in the case studies in Section 4 below.

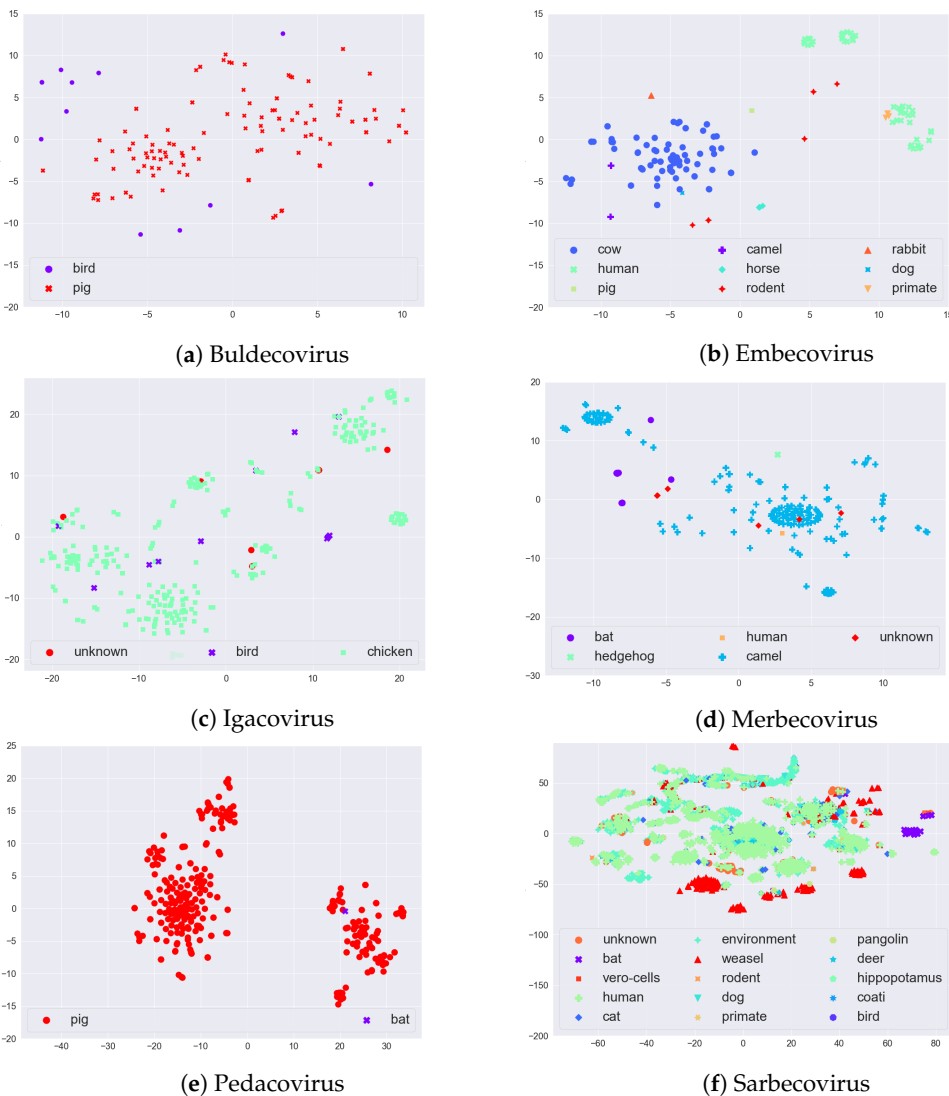

**Figure 3.** t-SNE plots for the six major subgenera (as mentioned in Table 2) of the Coronaviridae, colored by host.

## 3. Information Gain and Solvent Accessibility

For each major subgenus (of Table 2), we computed the information gain (IG) of each site in its alignment, according to Equation (1), and extracted the solvent-accessible positions. We achieved this using the SCRATCH Protein Predictor, which uses ACCpro for predicting the relative solvent accessibility of protein residues [25–27]. The ACCpro

predictor is based on one-dimensional recursive neural networks (1D-RNNs), with each amino acid residue predicted as buried or exposed from a scale of 1 to 20.

Given the solvent accessibility and IG for each site of a spike protein of a coronavirus, the Spearman rank correlation coefficient ($r_s$) and Kendall rank correlation coefficient ($\tau$) were then calculated for each subgenus according to the following definitions.

**Definition 2** (Spearman Rank Correlation). *The Spearman rank correlation coefficient $r_s$ measures the degree of association between two variables, where both variables are ordinal. The $r_s$ can be calculated as:*

$$r_s = 1 - \frac{6 \times \Sigma(d_i^2)}{n \times (n^2 - 1)} \quad (2)$$

*where $d_i$ is the difference in the rank of $S_i$ and $IG_i$, and n is the number of observations.*

**Definition 3** (Kendall Rank Correlation). *The Kendall rank correlation coefficient $\tau$ is a measure of the concordance between two variables, where both variables are ordinal or categorical. The $\tau$ can be calculated as:*

$$\tau = \frac{c - d}{n \times \frac{(n-1)}{2}} \quad (3)$$

*where c is the number of concordant pairs, d is the number of discordant pairs, and n is the number of observations.*

The values of these coefficients range from $-1$ to 1, where $-1$ represents a perfect negative correlation, 0 represents no correlation, and 1 represents a perfect positive correlation. The rank correlation between IG values and the average rank of each site is given in Table 6 using the Spearman and Kendall rank correlation.

The results show that there is a moderate to strong positive correlation between the solvent accessibility of different sites on the spike protein and their information gain for each subgenera. This implies that the sites of the spike protein that are more solvent accessible (i.e., exposed) also have higher information gain, indicating that they are more informative about the host. The *p*-value of all of the viruses is very low, which means that the correlation is statistically significant at the level of 0.05 or less, indicating that the observed correlation is unlikely to be due to chance. The notably strongest correlation is viewed in Sarbecovirus, which has many more hosts than the other subgenera, which is likely due to its disproportionately many ($\approx$8K) sequences compared to the other subgenera (due to the fact that it contains SARS-CoV-2); hence, we would expect the same trend in a stronger correlation as the number of sequences of subgenera increases.

The information gain of a site can be interpreted as the reduction in uncertainty or disorder of the host variable given that site. Therefore, if a site has a high information gain, it means that it provides a large amount of information about the host, and if a site has a low information gain, it means that it provides very little information about the host. Hence, since we can observe in Table 6 that solvent accessibility is positively correlated with information gain, this implies that the sites of the spike protein that are more solvent-accessible (i.e., exposed) are also more informative about the host; that is, IG can be used as a proxy for solvent accessibility. This is desirable, given that it is much more computationally expensive (in terms of runtime and memory usage) to infer solvent accessibility (with SCRATCH) than to compute information gain (see Table 7).

Although the study of solvent exposure comes with many advantages, such as providing direct information about the exposure of residues in the spike protein to the outside environment, which can directly impact its stability, function, and interactions with other molecules, its main disadvantage is the time it takes to compute its values when they are not known previously (see Table 7). Now that the number of sequences for viruses such as the Coronaviridae is larger, we can use the much more easy-to-compute information gain as a proxy. While, e.g., SARS-CoV-2, is a fairly well-studied virus by now, much is known about the spike protein structure and its solvent accessibility [16]; however, this proof of

concept that information gain can be used as a proxy for solvent accessibility could be used for other lesser-known families of viruses, where one would need to first compute or infer solvent accessibility.

**Table 7.** Resource allocation comparison. The memory usage for IG is negligible, and hence not reported.

| | SCRATCH | | Information Gain |
|---|---|---|---|
| | **Avg. $\pm$ Std. Runtime (s)** | **Avg. $\pm$ Std. Memory (kb)** | **Avg. $\pm$ Std. Runtime (s)** |
| Buldecovirus | 596.77 $\pm$ 115.03 | 20,217,835.84 $\pm$ 2098.98 | 0.08 |
| Igacovirus | 655.56 $\pm$ 143.94 | 20,220,781.72 $\pm$ 2050.28 | 0.13 |
| Embecovirus | 1209.55 $\pm$ 149.92 | 20,199,994.43 $\pm$ 3598.82 | 0.09 |
| Merbecovirus | 1123.03 $\pm$ 170.79 | 20,217,275.71 $\pm$ 2520.71 | 0.11 |
| Pedacovirus | 1223.19 $\pm$ 112.75 | 20,237,607.46 $\pm$ 2847.79 | 0.15 |
| Sarbecovirus | 1004.81 $\pm$ 186.91 | 20,213,971.42 $\pm$ 310,471.89 | 0.21 |

## 4. Case Studies

While the main result of this work is the connection between information gain and solvent accessibility, detailed in the previous Section 3, we provide a few case studies in the following. Since the Coronaviridae are fairly well-known now, due to the recent in-depth research on SARS-CoV-2, for example, such a study is more of a proof of concept—something that would certainly be useful for discovering new information about lesser-known families of viruses. To further validate this proof of concept, we explored a few case studies of the Coronaviridae regarding how some of the known biological evidence relates to the results we obtained.

### 4.1. Region-Wise Analysis

The comparison of normalized values of IG and SCRATCH is shown in Figure 4 for the six major subgenera of the Coronaviridae (see Table 2). Region-wise, the spike protein of roughly 1630 amino acids is divided into S1 and S2 regions, where the S1 domain ends (the S2 domain starts) roughly at position 770. Bashor et al., in [2], studied SARS-CoV-2 evolution in animals and showed that substitutions in spike proteins that include H69, N501, and D614, which also vary in human lineages of concern, were identified in non-human hosts, including dogs, cats, and hamsters. The complete list of mutations (as reported by Bashor et al. in [2]) in different hosts is reported in Table 8. Since we have cats as a host in Sarbecovirus (see the t-SNE plot in Figure 3f), we can observe in Figure 4f and Table 8 that the IG and SCRATCH values for positions 69 and 614 are on the moderate to low side, while position 501 has a high SCRATCH value, but a low IG, which could be due to the behavior of other hosts. Interestingly, according to the Centers for Disease Control and Prevention (CDC) website, some of the key mutations that have been reported in the spike protein of SARS-CoV-2, the virus that causes COVID-19 in humans, include N501Y and D614G (https://www.cdc.gov/coronavirus/2019-ncov/variants/variant-info.html) (accessed on 9 February 2022).Trends in Figure 4 that remain to be explored are the relatively low information gain (IG) in the last region (starting at position 1500) in Buldecovirus, Embecovirus, and Merbecovirus, and the relatively high IG in the last region of Pedacovirus. Other trends remain to be identified and explored.

Overall, we can observe that there are specific mutation positions within the spike protein that are important in terms of coronavirus detection as determined by biologists. Using tools such as SCRATCH values or IG can identify those possible mutations very quickly and efficiently, which can help us to understand the virus behavior in different hosts.

**Table 8.** Mutations at specific sites and the respective (normalized) IG and SCRATCH values for different species in Sarbecovirus, as reported by Bashor et al. in [2].

| Variant | Normalized IG | Normalized SCRATCH |
|---------|---------------|--------------------|
| D614G | 0.412017875 | 0.114198229 |
| H655Y | 0.009924703 | 0.162774760 |
| H69R | 0.090512449 | 0.154915497 |
| D138Y | 0.272433634 | 0.198103717 |
| D215N | 0.342889516 | 0.496680247 |
| N501T | 0.028416994 | 0.818966601 |
| S686G | 0.029752971 | 0.074078266 |

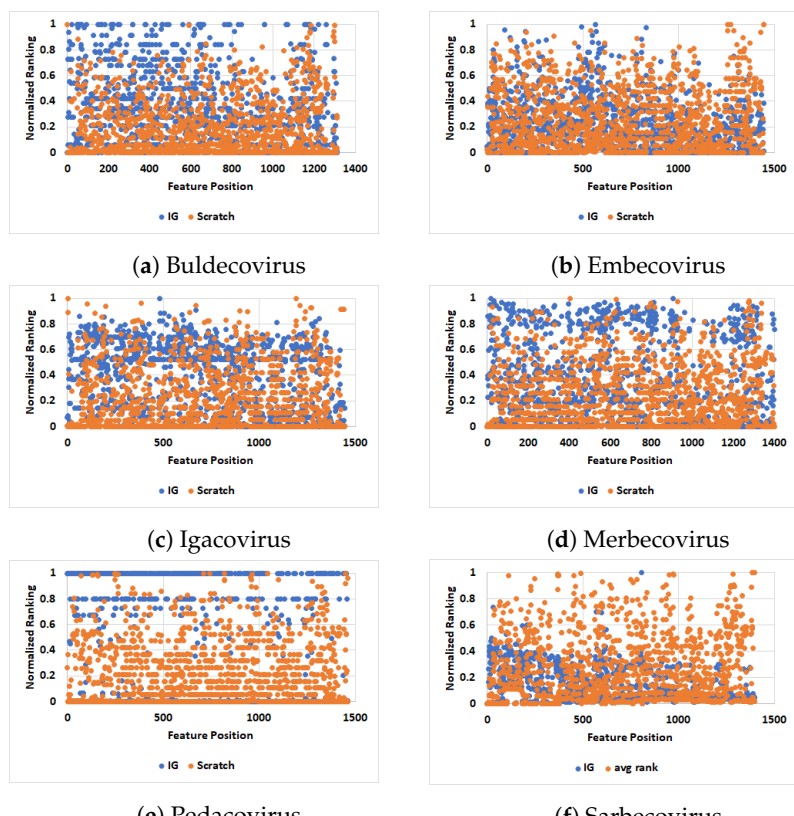

(**a**) Buldecovirus  (**b**) Embecovirus

(**c**) Igacovirus  (**d**) Merbecovirus

(**e**) Pedacovirus  (**f**) Sarbecovirus

**Figure 4.** Scatter-plot-based comparison of different viruses for the normalized values computed using SCRATCH and IG. The figure is best seen in color.

*4.2. Pairwise Studies*

Based on the literature and some of the trends given in the t-SNE plots of Figure 3, we focused here on a few pairs of hosts of interest. In particular, we paired up different hosts with Human to see which sites have the highest information gain. We took combinations such as Bat–Human, Cat–Human, Deer–Human, and Weasel–Human from Sarbecovirus, and Cow–Human from Embecovirus. In each case, we took all sequences from the host with the smaller set of sequences, paired it up with an equal number of randomly selected sequences for the other host, and multiply aligned them with Mafft (using default parameters). Statistics on the average sequence length, standard deviation (SD) of the length distribution, and alignment template length (similar to Section 2.2) are given in Table 9. We then computed information gain (IG) for each site in the respective alignment. The full list of amino acid sites and corresponding IG values for the pairs of hosts is available online (https://github.com/sarwanpasha/Comparative_Genomics/tree/main/Information_Gain/Set_of_Hosts) (accessed on 9 February 2022). What is interesting is that sequence variability (as determined by the t-SNE plots) is not always concordant with information

gain (as seen here in the averages). The point of studying information gain (or solvent accessibility) is based on the idea that sites contribute non-uniformly to host specificity, as seen in Figure 4.

**Table 9.** Length statistics for the different host pairings of this case study.

| Host Pair | Subgenus | No. Seq. | Avg. Len. | SD | Align. Len. | Avg. IG |
|---|---|---|---|---|---|---|
| Bat–Human | Sarbecovirus | 238 | 1257.81 | 14.49 | 1308 | 0.1914 |
| Cat–Human | Sarbecovirus | 550 | 1271.48 | 1.32 | 1279 | 0.0070 |
| Deer–Human | Sarbecovirus | 294 | 1271.19 | 8.47 | 1276 | 0.0114 |
| Weasel–Human | Sarbecovirus | 2378 | 1271.56 | 1.20 | 1279 | 0.0104 |
| Cow–Human | Embecovirus | 96 | 1359.95 | 3.96 | 1402 | 0.1445 |

## 5. Conclusions

In this work, we studied information gain (IG) as a source of information to explain host specificity. Such an approach is flexible to the continuously updating sequence information since it just requires a multiple-sequence alignment. We showed a connection between high IG and solvent accessibility, suggesting that proteins exposed to (solvents in) the outside environment are more responsible for host specificity. We also performed a visualization of the sequences to see some trends between different hosts in a given viral subgenus, performing some case studies on some regions and pairs of hosts of interest.

Future work includes more measures of host specificity beyond solvent accessibility. Since we considered spike protein (amino acid) sequences in this study, we could not perform this; however, if given nucleotide sequences, it would be interesting to see if there are sites that are positively or negatively selected (via an analysis of, e.g., dN/dS) for IG, solvent accessibility, etc. Finally, some subgenera have many more sequences or a wider diversity of hosts than others. Exploring the effects of this, but also of imperfect labeling (e.g., many viral sequences have a spectrum of hosts, even though only one is specified in the data), is another future direction—perhaps a more phylogeny-aware analysis could correct for some of this. Connecting other proteomic features or aspects of the protein structure, such as a secondary and possibly tertiary structure, is another interesting line of future work. Finally, using this idea to obtain domain knowledge for other studies, as a basis for further investigation, or to improve the performance of machine learning models could be other interesting future directions.

**Author Contributions:** Conceptualization: M.P.; Methodology: S.A., B.B.; Validation: M.P.; Formal Analysis: S.A.; Investigation: B.B.; Data Curation: M.P., S.A.; Writing—original draft preparation: S.A., B.B., M.P.; Writing—review and editing: M.P., S.A.; Visualization: S.A.; Supervision: M.P.; Project administration: S.A. All authors have read and agreed to the published version of the manuscript.

**Funding:** This research is supported by an MBD Fellowship for S.A. and a Georgia State University department of Computer Science start-up grant for M.P.

**Institutional Review Board Statement:** Not applicable.

**Informed Consent Statement:** Not applicable.

**Data Availability Statement:** Data is available at https://github.com/sarwanpasha/Comparative_Genomics.

**Conflicts of Interest:** The authors declare no conflict of interest.

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
