# Peer review of "Solvent Accessibility of Coronaviridae Spike Proteins through the Lens of Information Gain"

_2571-8800, doi:10.3390/j6020018_

Round 1

Reviewer 1 Report

Please see the attached document for the review report.

Author Response

Please see our response in the attached pdf

Author Response

(The authors gave the same response as above.)

Reviewer 3 Report

·         In line 122: what “Combining” means, there is no numbering, and the word followed by full stop.

·         In line 127, it is not scientifically to mention “we” in the text: “We then considered only the “major” subgenera and hosts”. Please state the sentence in passive.

·         The titles of tables 2 and 3 are not sufficient

·         Please mention the reference of the data in tables 2 and 3.

·         Also, the titles of Tables 4 and 5 are not sufficient

·         In figure 2, please provide more information about the data obtained in this figure, illustrate more about genera and subgenus.

·         In figure 3, state that “as mentioned in table 2”, not “Table 2” only.

·         In all tables, all the titles should be above the tables not beneath.

·         In figure 4, is it correct to say “The Figure is best seen in color” in the figure title?

·         The number of references is too little, only 16.

·         What is the importance of section 4 “case study”?

Author Response

(The authors gave the same response as above.)

Round 2

Reviewer 1 Report

please find the recommendations in the attachment.

Reviewer 3 Report

thanks a lot for clarifying the points mentioned in the revision